# Epigenetic Insights on PARP-1 Activity in Cancer Therapy

**DOI:** 10.3390/cancers15010006

**Published:** 2022-12-20

**Authors:** Giulia Pinton, Sara Boumya, Maria Rosa Ciriolo, Fabio Ciccarone

**Affiliations:** 1Department of Pharmaceutical Sciences, University of Piemonte Orientale (UPO), 28100 Novara, Italy; 2Department of Biology, University of Rome “Tor Vergata”, 00133 Rome, Italy; 3IRCCS San Raffaele Roma, 00166 Rome, Italy

**Keywords:** poly(ADP-ribosyl)ation, DNA methylation, histone modification, PARP inhibitors

## Abstract

**Simple Summary:**

PARP-1 and poly(ADP-ribosyl)ation control gene expression, DNA repair pathways, and genomic stability in multiple ways, such as affecting chromatin remodelling. This review article summarises how PARP-1 activity directly modifies histone proteins and the enzymes involved in DNA/histone epigenetic modifications to mould chromatin structure during transcription and DNA damage response. Understanding the role of poly(ADP-ribosyl)ation in the epigenetic regulation of chromatin organisation will help clarify resistance mechanisms to PARP inhibitors and highlight the clinical relevance of a combinatory approach based on epigenetic drugs.

**Abstract:**

The regulation of chromatin state and histone protein eviction have been proven essential during transcription and DNA repair. Poly(ADP-ribose) (PAR) polymerase 1 (PARP-1) and poly(ADP-ribosyl)ation (PARylation) are crucial mediators of these processes by affecting DNA/histone epigenetic events. DNA methylation/hydroxymethylation patterns and histone modifications are established by mutual coordination between all epigenetic modifiers. This review will focus on histones and DNA/histone epigenetic machinery that are direct targets of PARP-1 activity by covalent and non-covalent PARylation. The effects of these modifications on the activity/recruitment of epigenetic enzymes at DNA damage sites or gene regulatory regions will be outlined. Furthermore, based on the achievements made to the present, we will discuss the potential application of epigenetic-based therapy as a novel strategy for boosting the success of PARP inhibitors, improving cell sensitivity or overcoming drug resistance.

## 1. Introduction

ADP-ribosylation is catalysed by ADP-ribosyltransferase enzymes belonging to the Poly(ADP-ribose) polymerase (PARP) family and represents one of the most complex protein post-translational modifications. The PARP family is composed of 17 members, which are very heterogeneous from each other in terms of size (from about 36 kDa of PARP-16 to more than 200 kDa of PARP-14), localisation (nucleus, cytoplasm, mitochondria), and, above all, mechanisms and products of catalysis. In detail, the actual PARP enzymes that catalyse poly(ADP-ribosyl)ation (PARylation) producing polymers of ADP-ribose (PAR) are PARP-1 (the founding member of the family), PARP-2 (showing redundant functions with PARP-1), PARP-5a, and PARP-5b (also known as tankyrases due to the presence of the ankyrin domain involved in recognition of specific protein targets). Excluding two inactive enzymes (PARP-9 and PARP-13), the remaining members, and thus the majority of PARP family enzymes, are mono(ADP-ribosyl) transferases (MARTs) catalysing the addition of a single ADP-ribose unit onto target proteins. Notably, MARTs also include two members, PARP-3 and PARP-4, bearing the catalytic triad H-Y-E classically associated with PARylating activity, confirming the complexity of the catalytic mechanism of the PARP family. PARP-3 is historically associated with PARP-1 and PARP-2 for their ability to be activated by DNA breaks; PARP-4 is the most peculiar enzyme of the family, as it is the only one that has the catalytic domain in a central position of the structure and not at the C-terminus and because it can also catalyse PARylation, but only when localised to vault ribonucleoprotein particles. This heterogeneity in the composition of the PARP family accounts for the pleiotropic functions played by PARP/MART enzymes including, from a cellular point of view, the control of genome stability and transcriptional/post-transcriptional machinery or, from a broader systemic perspective, the regulation of inflammation and metabolism [1,2,3,4].

PARP-1 is the most abundant and characterised enzyme of the PARP family, responsible for most PAR chains synthesised in mammalian cells [5]. Although mitochondrial localisation has also been described [6], PARP-1 is mainly localised in the nucleus, where it is fundamental for the regulation of DNA repair. The recognition of DNA single- and double-strand breaks (SSB and DSB) stimulates PARP-1 activity following exogenous insults (*e.g.*, genotoxic agents) or when they originate from endogenous processes such as following endonuclease action during DNA repair. For this reason, PARP-1 participates in base excision repair (BER), nucleotide excision repair (NER), homologous recombination (HR), and nonhomologous end-joining (NHEJ) [1,5].

Several pre-clinical approaches have been tested to inhibit PARP-1 activity in combination with DNA-damaging agents to limit cell damage recovery and improve chemotherapeutic efficacy. More importantly, PARP inhibitors (PARPi) revolutionised the field of precision medicine in cancer when they were demonstrated to selectively kill tumours bearing germline mutations in the HR-associated genes Breast Cancer gene 1 and 2 (*BRCA1/2*) [7,8]. This process represents a genetic concept known as synthetic lethality. The abrogation of PARP activity in cells with defective HR pathways leads to the accumulation of harmful unrepaired DNA breaks that ultimately induce cytotoxicity. FDA-approved PARPi (olaparib, niraparib, rucaparib, and talazoparib) are the first synthetic lethal targeted drugs to enter clinical practice and are currently used to treat ovarian, breast, pancreatic, or prostate cancer characterized by *BRCA1/2* mutations [9].

Unfortunately, the efficacy of PARPi is rapidly blunted by drug resistance mechanisms that can avert the toxic persistence of PARP-1 at DNA breaks or reactivate HR pathways [5,10]. The constant presence of PARP-1 in damaged chromatin, known as “PARP trapping,” is recognised as the leading cause of PARPi-induced cytotoxicity in cells with a defective HR system due to replication fork collapse and accumulation of unrepaired DNA DSB [11,12]. Therefore, any event counteracting PARP trapping can induce PARPi resistance as determined by the acquired mutation R591C in the PARP-1 protein [13]. Secondary somatic mutations able to overcome HR defects are among the most common causes of PARPi resistance. Genetic events that directly restore BRCA1/2 function include the reversion of the inherited genetic defect leading to the expression of wild-type proteins, mutations that can overcome frameshift defects producing nearly full-length proteins, and splice variants codifying for hypomorphic truncated proteins [10,13,14]. Alternatively, PARPi resistance can be determined by mutations in genes that restore HR regardless of BRCA1/2 function as observed by loss of function mutations in the *TP53BP1* gene [10,15].

During the last few decades, PARP-1 has emerged as a critical regulator of chromatin dynamics associated with DNA damage response (DDR) and transcriptional regulation by affecting epigenetic mechanisms at different levels [16,17]. In this review, we will outline how PARylation directly modifies DNA and histone epigenetic modifiers and how epigenetic drugs confer sensitivity or help to overcome resistance to PARPi in a combinatory approach.

## 2. Background Knowledge of PARP Reactions

PARP/MART enzymes catalyse the transfer of ADP-ribose moieties from nicotinamide adenine dinucleotide (NAD^+^) to specific amino acid residues of target proteins. Once the first ADP-ribose monomer has been attached to the protein, PARP enzymes proceed with the elongation reaction, catalysing ribose–ribose glycosidic bonds between ADP-ribose and introducing branching points along the PAR chain at intervals of about 25 units (Figure 1A). Therefore, the PARylation reaction allows the generation of long PAR chains, linear or branched, that can also reach more than 200 units [1,2,4].

PAR turnover is very rapid and achieved by the catabolic activity of several PAR-degrading enzymes. Poly(ADP-ribose) glycohydrolase (PARG) was considered the only enzyme with PAR hydrolase activity for a long time. PARG leads to the cleavage of ribose–ribose bonds without removing the terminal ADP-ribose unit from the substrate. The complete degradation of PAR modification is accomplished by the terminal ADP-ribose protein glycohydrolase (TARG1)/C6orf130, MACRO domain containing 1 (MacroD1), and MacroD2 that hydrolyse the last ADP-ribose moiety attached to the proteins. In addition, ADP-ribosyl hydrolase 3 (ARH3) is likely the sole enzyme having the same endo- and exoglycosidic activities as PARG but also acting on the last unit [2,3].

### 2.1. Covalent PARylation

Covalent PARylation of proteins was historically believed to occur at glutamate and aspartate residues. However, more recent mass spectrometry techniques have identified several other amino acids as PARylation targets, including lysine, tyrosine, and, above all, serine [18,19,20,21]. More specifically, histone PARylation factor 1 (HPF1) interacts with PARP-1 and PARP-2 and provides their catalytic domains with an additional catalytic residue that determines the preferential PARylation of serine [22].

PARP-1 (and PARP-2) proteins possess several amino acid residues that undergo PARylation and are all contained in the so-called automodification domain (Figure 1B). PARP-1 automodification is highly induced upon recognising DNA breaks to permit PARP-1 dissociation from the DNA damage sites, avoiding PARP-1 trapping [5,18]. Three serine residues (S499, S507, and S519) were recently demonstrated as the main targets of PARP-1 automodification *in vivo* and, when mutated, they impede PARP-1 delocalisation from DNA lesions and sensitise cells to PARPi [23]. This evidence further confirms the relevance of PARP-1 trapping in inducing cytotoxicity.

### 2.2. Non-Covalent PARylation

PARP-1 automodification substantially contributes to the pleiotropic role of the enzyme, expanding the number of molecular factors that can interact with it. Consistently, proteins bearing specific amino acid motifs (or even structural domains) can bind PAR chains non-covalently with different dissociation constants that determine high, medium, and low affinities of interaction. In this way, automodified PARP-1 functions as a molecular platform for recruiting factors that must attend to a specific process, such as DNA repair. The most widespread PAR-binding module is the classical “PAR-binding motif” (PBM), which mainly consists of about 20 hydrophobic and basic amino acid residues, the positive charge of which is likely to determine the electrostatic affinity to negatively charged PAR chains. Other PAR-interacting modules include the PAR-binding zinc finger (PBZ), containing two conserved cysteines and two histidines; the macrodomain, which is a well-defined protein domain of about 100–200 amino acids found in PAR-hydrolysing enzymes (PARG, MacroD1/D2, and TARG1) and in several chromatin proteins. Different types of DNA- and RNA-binding motifs are likely to interact with PAR due to the similarity they have with nucleic acids [24,25].

## 3. PARP-1 Activity Shapes Chromatin during DNA Repair and Transcription

The ability of PARP-1 to act as a sensor of DNA damage is guaranteed by the widespread distribution of the enzyme onto chromatin. In the inactive state PARP-1 binds nucleosomes, promoting chromatin condensation [16,26]. The induction of PARP-1 activity by DNA breaks locally and transiently relaxes chromatin, reducing the affinity of PARP-1 for nucleosomes and facilitating the recruitment of DNA repair factors that must operate in those sites [16,27]. The same mechanism is required for PARP-mediated regulation of transcription [17,28,29]. In this context, PARP-1 contributes to the long-term chromatin relaxation required for sustained transcription. The histone H4 was identified as a local signal promoting PARP-1 activity [30]. Other chromatin-recruited factors, such as transcription factors or RNA molecules, have been described to stimulate PARP-1 activity independently from DNA damage and may locally trigger chromatin loosening via PARP-1 activation [5,31].

The mechanisms of activation of PARP-1 during DDR and transcriptional activation are distinct. The recognition of DNA breaks occurs via the Zn finger domains contained in the N-terminal DNA binding domain (DBD) of PARP-1. They cooperatively interact with the tryptophan–glycine–arginine-rich (WGR) and catalytic domains at the C-terminus to trigger the rapid and transient activation of PARP-1 necessary to recover the lesions [27,32]. On the other hand, the transcription-associated long-term activation of PARP-1 entails the stimulatory interaction of the PARP-1 catalytic domain with nucleosomes that expose histone H4 [30] (Figure 1B). In this context, the DBD domain of PARP-1 permits the nonspecific binding of DNA along chromatin to scan the genome and reach regions that need to be decondensed. Consistently, PARP-1 is particularly enriched in promoter regions of actively transcribed genes where PARs assure an open chromatin configuration and the recruitment of transcription machinery [33,34].

The switch between condensed heterochromatin and relaxed euchromatic regions is strictly dependent on changes in DNA and histone epigenetic modifications. Growing evidence shows that the influence of PARylation on chromatin dynamics largely depends on changes in epigenetic patterns [5,31]. Hereafter, we will delineate the different levels of regulation that PARP-1 exerts on epigenetics: directly, by targeting DNA and histones; and indirectly, by modifying covalently and non-covalently epigenetic enzymes (Table 1). We will focus on DNA and histone epigenetic marks relevant for chromatin shaping and modulated by PARP-1 activity during DDR or transcriptional regulation.

## 4. PARP-1 and DNA Epigenetic Modifications

DNA methylation patterns are introduced by the action of DNA methyltransferases (DNMT) and removed by dilution of the 5-methylcytosine (5mC) mark during DNA replication (passive DNA demethylation) or by the hydroxylase activity of ten-eleven translocation (TET) family enzymes via the iterative production of 5-hydroxymethylcytosine (5hmC), 5-formylcytosine (5fC), and 5-carboxylcytosine (5caC) (active DNA demethylation). These oxidative intermediates of DNA methylation are removed by DNA repair mechanisms, primarily BER, but they can also act as stable epigenetic marks. The maintenance of DNA methylation assures genome stability while its dynamism promotes chromatin changes associated with DNA repair pathways and transcription [35].

### 4.1. DNA Methylation

The crosstalk between PARylation and DNA methylation has been mainly characterised in the context of gene expression. PARP-1 inhibition induces hypermethylation of methylation-free regulative regions at gene promoters (*e.g.*, CpG island) and imprinted loci [36,37,38,39]. The occupancy of these regions by PARP-1 supports the hypothesis that PARylation contributes to maintaining unmethylated states by abrogating DNA methyltransferase activity [40]. Consistently, DNMT1, which is typically involved in preserving DNA methylation patterns across cell divisions, can bind PARs non-covalently, and this interaction negatively affects its enzymatic activity [41]. No information is available about the effect of PARylation on the *de novo* DNA methyltransferase DNMT3A and DNMT3B.

Locus-specific inhibition of DNMT1 has been ascribed to the presence of automodified isoforms of PARP-1 or PARylated transcription factors [38,39,42,43]. Among the latter, a relevant role is played by the insulator factor CCCTC-binding factor (CTCF), which is a direct target of covalent and non-covalent PARylation [38,44]. Moreover, CTCF interaction with the PARP-1 protein stimulates its enzymatic activity independently of DNA damage [42,45]. Considering that CTCF preferentially recognises unmethylated DNA regions [44], the direct activation of PARP-1 represents one of the mechanisms limiting DNMT1 action at CTCF-bound regions. In support of this mechanism, the inhibition of PARP-1 activity leads to hypermethylation events at the promoter of *CDKN2A* (coding for the tumour suppressor p16^INK4A^) [43], normally preserved by PARylated CTCF, and at the unmethylated alleles of the imprinted locus *Igf2/H19* occupied by CTCF and automodified PARP-1 [38]. Further confirmation of a mutually exclusive localisation between PARP-1 and DNA methylation was obtained by genome-wide epigenetic analyses performed in breast cancer cell lines [46].

Nevertheless, a positive effect of PARylation on the hypermethylation of DNA regions has also been provided. In the context of ribosomal DNA (rDNA) silencing, PARP-1 activity is stimulated by RNA molecules to establish a silent hypermethylated state responsible for rRNA repression [47]. Whether the context-specific effect of PARP-1 on DNA methylation patterns may depend on the PARP-1 activators or locus-specific chromatin interactions needs to be evaluated.

### 4.2. DNA Demethylation and Hydroxymethylation

Besides impeding DNA hypermethylation in specific loci, persistent PARylation-dependent inhibition of DNMT1 can induce widespread passive DNA demethylation [37,40]. This phenomenon was observed following protracted hyperactivation of PARP-1 due to CTCF overexpression [45]. A contribution of PARylation on active DNA demethylation can also be envisaged, considering that PARP-1 activity can influence DNA hydroxylase enzymes, particularly TET1, at different levels. PARP-1 sustains TET1 expression and can modify it both covalently and non-covalently [39,48,49]. Merging results from *in vitro* analyses on PARylated TET1, the presence of PARs seems to negatively influence TET1 hydroxylase activity [48,49]. Moreover, TET-1 non-covalent PARylation was shown to mediate its recruitment at specific regulatory regions to promote local DNA demethylation [50]. Although alteration of 5hmC levels has been observed following PARP-1 inhibition, whether it can also influence the iterative conversion of 5hmC into 5fC/5caC has to be proven.

## 5. PARylation and Histone Modifications

DNA methylation patterns are orchestrated in concert with histone modification marks. Methylation and acetylation are the most studied modifications of core histone proteins (H2A, H2B, H3 and H4) in the context of transcriptional regulation. Still, many other modifications have also been identified, including ubiquitination, phosphorylation, SUMOylation, and PARylation itself [51]. As can be deduced from the variety of marks, the histone code is very complex. Thus, the epigenetic mechanisms leading to chromatin changes necessarily require hierarchical regulators involved in their coordination. The regulatory participation of PARylation in different epigenetic layers indicates PARP-1 as an ideal chromatin factor devoted to orchestrating epigenetic events. Moreover, histone PARylation code is starting to be defined, highlighting an intertwined relationship between PARs and other epigenetic marks.

### 5.1. Histone H1

The linker histone H1, one of the first identified PARylation targets in the native chromatin, contributes to high-order chromatin compaction by binding the linker DNA between nucleosomes. Following maximal activation of PARP-1 activity during DNA damage, H1 is covalently PARylated in different amino acid residues across the entire protein structure. It can also bind PARs non-covalently via the lysine-rich C-terminal domain [52,53]. *In vitro* and *in vivo* experiments mainly indicate that PARylation promotes H1 displacement from chromatin to promote chromatin relaxation, allowing the recruitment of DNA repair factors [53,54,55]. Interestingly, the crosstalk between H1 and PARP-1 is also relevant during the signalling process associated with DDR by regulating chromatin recruitment of ataxia telangiectasia mutated (ATM) kinase. The H1.2 isoform protects chromatin from aberrant ATM loading and activation in basal conditions. Upon DNA damage, PARP-1 covalently modifies H1.2 at the S188 residue, displacing it from chromatin to promote efficient recruitment and activation of ATM kinase at DNA damage sites [56] (Figure 2).

PARP-1-mediated displacement of H1 also controls gene expression, as observed for the transcriptional activation of progesterone-responsive genes or during the reprogramming of the neuronal gene network [57,58]. The mutual exclusion of H1 from PARP-1 bound regions has been demonstrated genome-wide, particularly at promoters of actively transcribed genes, which supports the idea that PARP-1 activity may influence gene expression by restraining H1-mediated chromatin condensation [59].

### 5.2. Histones H2A/H2B and Their Variants

All core histones can be extensively PARylated to promote chromatin relaxation during DNA repair and gene expression [60,61]. The specific PARylation of certain amino acids in histones can also induce regulatory effects on other epigenetic modifications or signalling processes. During adipogenesis, PARylation of histone H2BE35 by PARP-1 inhibits AMP kinase-mediated phosphorylation of the adjacent residue H2BS36, which is necessary to express pro-adipogenic genes [62].

In the family of histone H2A variants, H2AX acts as an acceptor of ADP-ribose during DDR. H2AX-E141 PARylation avoids DNA DSB formation and the associated signalling in the context of BER activation. In detail, PARylation of H2AX is stimulated by oxidative DNA damage and promotes BER via NEIL3 glycosylase recruitment at the sites of DNA breaks. At the same time, H2AX-E141 PARylation negatively regulates the phosphorylation of the adjacent S139 residue, thus limiting the activation of DSB repair mechanisms [63].

Several connections have been identified between MacroH2A1.1 and PARP-1 during DDR and transcriptional regulation due to the ability of this histone variant to bind PARs non-covalently. PARP-1 activity and MacroH2A1.1 have opposing effects on chromatin structure, decompacting and compacting chromatin, respectively. MacroH2A1.1 can also accomplish this thanks to its inhibitory effect on PARP-1 activity, thus contributing to the maintenance of chromatin plasticity in response to DNA damage [64]. By limiting the activation of PARP-1, MacroH2A1.1 can also induce gene silencing as observed at the *Hsp70.1* gene promoter, which instead becomes activated and PARylated following MacroH2A1.1 displacement triggered by heat shock [65]. Moreover, the recruitment of PARP-1 via MacroH2A1.1 binding is fundamental for regulating gene expression, either positively or negatively, and promoting H2BK12 and K120 acetylation at MacroH2A1-target genes [66]. Of note, the splice variant MacroH2A1.2 does not bind PAR directly, but it can compact chromatin through an activity situated in the linker region. However, it can co-localise with PARP-1 at DNA damage sites via the interaction with the histone demethylase KDM5A, which is recruited by binding PARs in a non-covalent manner. The PARP-1/MacroH2A1.2/KDM5A axis has been very recently described in the DDR, and its role deserves to be investigated in other chromatin contexts [67].

### 5.3. Histone H3

The roles of histone H3 epigenetic modifications are among the most characterised in the context of transcription-associated chromatin domains. Histone H3K9 trimethylation (H3K9me3) is coupled with high 5mC levels to promote chromatin condensation and transcriptional repression; on the other hand, regions devoid of DNA methylation are associated with gene silencing when enriched in H3K27me3 or with active transcription when occupied by H3K4me3 and H3 acetylation [68].

As described for other histone proteins, PARP-1 activity can modify histone H3 with direct consequences on other epigenetic marks. H3S10 represents the primary H3 amino acid targeted by PARP-1 *in vivo* during DDR [69]. H3S10 PARylation impairs the Aurora kinase-mediated phosphorylation of the same residue [70] and is mutually exclusive with H3K9 acetylation [21,71], without affecting the methylation at the same position [21]. Notably, the PARylation of histone H3 in the presence of DNA damage was also shown to reduce H3K27me3 levels due to the reduced affinity of the methyltransferase enhancer of zeste homolog 2 (EZH2) for its histone target. The H3 residue undergoing PARylation and affecting EZH2 recruitment has not been identified yet. We can hypothesise that PARylation of the same H3K27 or the adjacent H3S28 can accomplish this function. In fact, H3S28 is the only other serine of H3 known to be modified by PARP-1 upon DNA damage beyond H3S10 [69], and H3K27 is one of the most frequently PARylated H3 lysines [19]. Nevertheless, PARP-1 activity can also determine the decrease of H3K27me3 levels by covalently modifying EZH2, inducing the inhibition of its activity [72] or the dissociation from the polycomb repressive complex 2 (PRC2) with consequent proteasomal degradation [73].

While PARylation seems to limit EZH2 activity during DNA damage, it favours chromatin recruitment of members of the histone lysine demethylase (KDM) family. The KDM4D enzyme that typically demethylates H3K9me3/2 is recruited at DNA damage sites following covalent PARylation at the C-terminal domain. KDM4D recruitment promotes DSB repair pathways, facilitating ATM-mediated phosphorylation of its targets, including H2AX-S139 [74]. PARP-1 also leads to KDM5B recruitment via its covalent PARylation, enabling the removal of methyl groups from H3K4me3. H3K4me3 reduction at DNA lesions may also be achieved by recruiting the KDM5A enzyme via non-covalent PARylation [67]. Demethylation of H3K4me3 is critical for the action of proteins involved in the NHEJ or HR pathways [75].

Beyond loading DNA repair factors, such repressive epigenetic events are necessary to abrogate spurious transcription at DNA damage sites. Inhibitory effects on transcription have also been demonstrated after covalent PARylation of KDM4D and the histone methyltransferase nuclear receptor-binding SET domain 2 (NSD2). KDM4D PARylation at the N-terminal domain affects the removal of the repressive mark H3K9me2 at retinoic acid receptor-target genes [76]. At the same time, PARylated NSD2 is inhibited and no longer recruited at its target genes, impairing the demethylation of H3K36 [77]. PARP-1 activity maintains active transcription of gene promoters by covalently modifying either the histone demethylase KDM5B, in order to inhibit its activity and preserve H3K4me3 levels [34], or the acetyltransferase p300, to stimulate its activity and thus H3 acetylation [78].

**Table 1 cancers-15-00006-t001:** Epigenetic enzymes targeted by covalent or non-covalent PARylation.

Enzyme	Activity	PARylation	Outcome
DNMT1	DNA methyltransferase	Non-Covalent	Enzyme inhibition [41]
TET1/2	5mC hydroxylase	Covalent Non-Covalent	Enzyme inhibition [48,49]; recruitment at gene regulatory regions [50]
EZH2	H3K27 methyltransferase	Covalent	Enzyme inhibition [72]; dissociation from PRC2 complex [73]
KDM4D	H3K9me3/2 demethylase	Covalent	Recruitment at DNA damage sites [74]; putative enzyme inhibition [76]
KDM5A	H3K4me3 demethylase	Non-Covalent	Recruitment at DNA damage sites [67]
KDM5B	H3K4me3 demethylase	Covalent	Enzyme inhibition [34]; recruitment at DNA damage sites [75]
NSD2	H3K36 demethylase	Covalent	Enzyme inhibition and impairment of nucleosome binding [77]
p300	Histone acetyltransferase	Covalent	Enzyme activation [78]

### 5.4. Histone H4

The interplay between PARylation and the H4 histone is functionally relevant for the ability of H4 to stimulate prolonged PARP-1 activation in the context of gene expression regulation [30]. Moreover, residues of H4 have also been shown to take part in the histone code when PARylated. For instance, the PARylation of H4S1 can be introduced even in the presence of H4R3 dimethylation and H4K5 or H4K8 acetylation [21]. On the other hand, PARylation of H4K16 was shown to be introduced *in vitro* by PARP-1 activity but impaired by acetylation of the same residue [19]. It will be intriguing to verify whether H4K16 PARylation occurs *in vivo* during DDR and how the interplay with H4K16 acetylation occurs. H4K16 acetylation is fundamental for the decondensation of chromatin at DNA damage sites, as the acetyl mark abrogates nucleosome packaging by impairing the interaction of the H4 basic tail with a cluster of acidic residues present on an adjacent nucleosome [79,80].

## 6. Epigenetic Mechanisms in PARPi-Based Cancer Therapy

The use of PARPi in cancer therapy is an effective targeted approach in ovarian, breast, pancreatic, and prostate cancer subtypes with defective *BRCA1/2* genes, commonly identified as having the “BRCAness phenotype” [9]. Based on the pleiotropic effects of PARP-1 in the control of chromatin changes, several preclinical studies have investigated how the levels and activity of epigenetic enzymes can modify PARPi sensitivity and resistance. Ongoing clinical trials testing the efficacy of a combinatory approach of PARPi with epigenetic drugs in different cancer types highlight the current clinical interest in the cross-talk between epigenetics and PARylation (Table 2).

Mutations of DNA methylation/demethylation enzymes are common in haematological malignancies and mainly relate to *TET2* and *DNMT3A* genes. Recent data demonstrated that mutations of these two genes have opposite consequences on PARPi sensitivity in acute myeloid leukaemia (AML) cells expressing oncogenic tyrosine kinases (*e.g.*, mutated FLT3 or JAK2), which are characterised by high endogenous levels of DNA DSB. *TET2*-mutated AML cells have defective HR due to *BRCA1* and *LIG4* downregulation and tolerate spontaneous and drug-induced DNA damage through PARP-1-mediated alternative NHEJ. Therefore, this condition makes cells more vulnerable to PARPi. On the other hand, *DNMT3A*-mutated AML cells rely on HR and are insensitive to PARPi, exhibiting the downregulation of *PARP-1* gene expression [81]. The increased toxicity of PARPi in a *TET2*-mutated background was also ascribed to low levels of tyrosyl-DNA phosphodiesterase 1 (TDP1), which is known to repair topoisomerase I-induced DNA breaks in association with PARP-1. The generation of *TET2*-mutated AML cell lines revealed that TDP1 downregulation was due to reduced enrichment of 5hmC levels at the *TDP1* promoter in the presence of defective TET2. Nevertheless, no association between *TET2* mutations and reduced expression of *TDP1* was identified in human AML samples, confirming that this is not the only mechanism involved in the synthetic lethality of PARPi in *TET2*-mutated AML [82]. Beyond genetic alterations, TET genes are frequently deregulated in both solid and liquid tumours. A recent paper verified that TET1 upregulation in T-cell acute lymphoblastic leukaemia (T-ALL) increases global 5hmC levels regulating the expression of T-ALL oncogenes and genes involved in the cell cycle and DNA repair [83]. Based on the evidence that PARP-1 can impact the activity of the TET1 enzyme at the transcriptional and enzymatic levels [39,48,49], the use of the PARPi olaparib was effective in antagonising leukaemic growth of T-ALL, negatively impacting TET1 expression and 5hmC marks [83].

While conferring sensitivity to PARPi in AML, *TET2* loss contributed to PARPi resistance in *BRCA2*-deficient embryonic stem cells and mammary tumours. This effect was ascribed to the protection of stalled replication forks, a common mechanism involved in PARPi resistance, avoiding the recruitment of the BER enzyme APE1 at DNA damage sites. Promoting 5hmC formation allows the degradation of stalled replication forks, sensitising cells to PARPi and genotoxic agents [84]. Analogously, the EZH2-mediated accumulation of H3K27me3 at stalled replication forks is necessary to recruit the MUS81 repair enzyme. Therefore, loss of EZH2 is associated with the protection of stalled replication fork and resistance to PARPi in *BRCA2*-deficient cells [85]. Nevertheless, many papers also demonstrated that EZH2 inhibition enhances the efficacy of PARPi treatments, as observed in breast cancer [73] and CARM1-high ovarian cancer. In particular, the arginine methyltransferase CARM1 stabilises EZH2, which can repress NHEJ-related genes while activating HR repair. In this condition, the inhibition of EZH2 induces the switch from HR to NHEJ. Since the latter is an error-prone repair mechanism, it leads to genomic instability and cell death in combination with olaparib [86]. It should be noted that this effect of EZH2 inhibitors is operative in HR-proficient cells, suggesting their potential use in PARPi-resistant tumours that reactivate the HR pathway.

Another strategy that can increase PARPi sensitivity in HR-proficient cells is the induction of the BRCAness phenotype by repressing wild-type *BRCA1/2* genes or other relevant HR-related factors. This event was shown to occur after treatment with several types of epigenetic drugs, such as inhibitors of histone deacetylases (HDACi), which are known to repress several HR enzymes in breast cancer [87], and iron chelators, such as VLX600, which inhibit KDM enzymes with consequent accumulation of H3K9me3, disrupting the recruitment of HR repair proteins to DNA DSB [88]. The BRCAness phenotype can also be induced in non-small cell lung cancer (NSCLC) by using the DNA methyltransferase inhibitor (DNMTi) 5-azacytidine (5-AZA). Low doses of 5-AZA cause the downregulation of genes involved in HR and NHEJ, sensitising cells to the PARPi talazoparib. This condition also exacerbates PARP-1 trapping at DNA damage sites and the cytotoxic accumulation of DSBs. Nevertheless, the primary mechanism responsible for cell death is the reprogramming of DNA repair pathways. In fact, the use of veliparib, a catalytic inhibitor of PARP-1 unable to efficiently induce PARP-1 trapping, is still effective in combination with 5-AZA [89]. Similar results on the efficacy of PARPi and hypomethylating agents were also demonstrated in AML and breast cancer [90,91]. Furthermore, different types of PARPi showed promising therapeutic effects for treating arsenic trioxide-acute promyelocytic leukaemia (APL) in combination with DNMTi or high-dose ascorbate that is shown to increase 5hmC and thus the DNA demethylating ability of TET enzymes [92].

## 7. Conclusions

PARP-1 is typically associated with DDR by the vast majority of researchers or, even worse, it is only considered a marker of apoptotic cell death. Nevertheless, the functional importance of PARP-1 and PARylation in basal and stimulus-induced transcription, chromatin insulation, and nuclear architecture organisation nowadays is a fact. All these processes necessitate a fine-tuned regulation of chromatin structure, and PARP-1 can impinge on it in different ways. PARP-1 can promote chromatin condensation independently of its enzymatic activity or it can shape euchromatic and heterochromatic regions, directly affecting epigenetic machinery, the histone code, and DNA methylation patterns. More recently, DNA PARylation was also identified in mammalian cells at adenosine residues in single-stranded DNA regions. This DNA modification is introduced by PARP-1 and is not associated with DNA damage. Future studies must uncover if DNA PARylation has functional roles in chromatin regulation at genomic regions characterised by localised DNA structures that physiologically contain single-stranded DNA, such as R-loops and G-quadruplex DNA [93].

The different layers of regulation that PARP-1 activity exerts on histone/DNA epigenetic modifications are certainly involved in the success or failure of PARPi in cancer therapy. Beyond *TET2* and *DNMT3A*, mutations or amplification of genes codifying for other epigenetic enzymes, such as the histone demethylases KDM5A and 6A [94], are frequently observed in cancer and may be assessed for their synthetic lethality with PARPi to identify new patients cohorts eligible for PARPi targeted therapy. Combinatory approaches of PARPi with epigenetic drugs seem to have success mainly when the impinged epigenetic pathways can induce the BRCAness phenotype. This effect can be ascribed to deregulated chromatin rearrangements at DNA damage sites that affect the efficient recruitment of DNA repair factors or transcriptional changes undermining cell homeostasis. Although bromodomain and extraterminal domain (BET) proteins were not described in our review because they are not epigenetic modifiers but epigenetic readers involved in transcriptional regulation, it is important to highlight that BET inhibitors can sensitize *BRCA*-proficient tumours to PARPi by downregulating several HR-related factors, including BRCA1, thus inducing the BRCAness phenotype [95,96,97].

The acquisition of resistance is the main factor limiting the efficacy of PARPi. In addition to acquired mutations, epigenetic reprogramming and chromatin remodelling elicited by drug pressure establish new transcriptional networks that can induce the resistant phenotype. Based on the pleiotropic epigenetic action of PARP-1, whether the long-term effects of PARPi on DNA and histone epigenetic patterns may directly influence therapeutic success has to be verified. New insights on epigenetic rearrangements in clinical specimens following PARPi treatments can provide remarkable clues about using epigenetic agents in combination with PARPi to limit or overcome resistance.

## Figures and Tables

**Figure 1 cancers-15-00006-f001:**
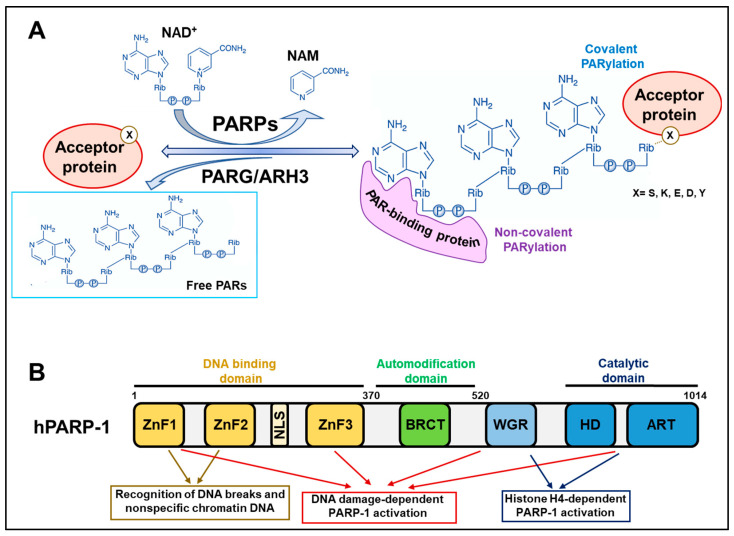
Structural and functional characteristics of PARP-1. (**A**) PARP activity mediates covalent PARylation, catalysing the transfer of ADP-ribose moieties from NAD^+^ to several amino acid residues (X) on acceptor proteins releasing nicotinamide (NAM). The reversal reaction is catalysed by PAR-hydrolysing enzymes, including PARG/ARH3. Different protein motifs or domains can recognise and bind PARs non-covalently. (**B**) Schematic representation of the human PARP-1 molecular structure. The PARP-1 structure can be subdivided into three major domains: the NH_2_-terminal DNA binding domain, the central automodification domain, and the COOH-terminal catalytic domain. Other subdomains or motifs can be identified in the hPARP-1 structure: ZnF (zinc finger domain), NLS (nuclear localisation signal), BRCT (BRCA1 C-terminal domain), WGR (Trp-Gly-Arg domain), HD (helical domain), and ART (ADP-ribosyl transferases domain).

**Figure 2 cancers-15-00006-f002:**
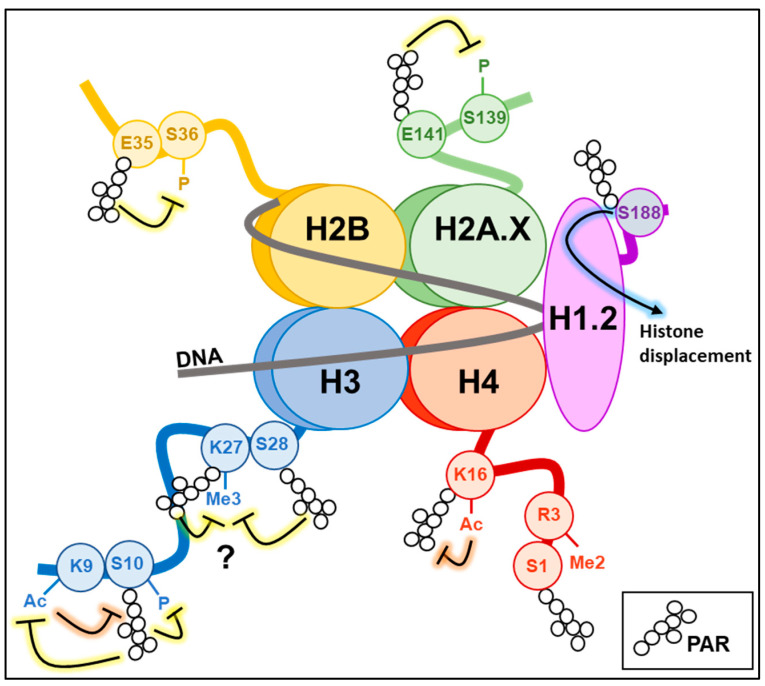
Histone PARylation code. The picture shows the PARylation of amino acid residues identified *in vivo* or *in vitro* on histone tails and their relative effects (known or hypothesised) on other histone epigenetic marks. Ac: acetylation; Me3: trimethylation; P: phosphorylation.

**Table 2 cancers-15-00006-t002:** Ongoing clinical trials of PARPi in combination with epigenetic drugs.

NCT Identifier (Estimated Study Completion Date)	Cancer Type	Interventions	Phase	Output
NCT 02878785 (December 2022)	Acute Myeloid Leukaemia	PARPi: talazoparib DNMTi: decitabine	1 2	Dose finding based on tolerability, efficacy, and pharmacodynamic data Efficacy of the selected combination regimen
NCT 04846478 (September 2023)	Metastatic castration-resistant prostate cancer	PARPi: talazoparib EZH2i: tazemetostat	1	Safety, tolerability, and preliminary clinical activity of drug combination
NCT 03742245 (September 2024)	Relapsed/refractory and/or metastatic breast cancer	PARPi: olaparib HDACi: vorinostat	1	Safety and preliminary efficacy of drug combination
NCT 04355858 (April 2025)	HR+/HER2- endocrine-resistant advanced breast cancer	PARPi: SHR3162 EZH2i: SHR2554	2	Screening valuable treatment cohorts for randomized controlled phase III clinical studies with larger sample size
NCT 05071937 (November 2027)	Recurrent ovarian, fallopian tube, or primary peritoneal carcinoma	PARPi: talazoparib BETi: ZEN003696	2	Efficacy of drug combination

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
