# Peer review of "Epigenetic Insights on PARP-1 Activity in Cancer Therapy"

_cancers, 2022, doi:10.3390/cancers15010006_

Round 1

Reviewer 1 Report

 Article: Epigenetic insights on PARP-1 activity in cancer therapy

Authors: Giulia Pinton, Sara Boumya, Maria Rosa Ciriolo and Fabio Ciccarone

Manuscript Number: cancers-2059487

As stated, this review is focusing on the implication of PARP-1 and PARylation in controlling epigenetic events involved in chromatin changes occurring during DNA repair and gene transcription. Authors discussed the potential application of epigenetic-based therapy as a novel strategy for boosting the success of PARP inhibitors, improving cell sensitivity or overcoming drug resistance. 

In my opinion, the review paper meets criteria for publication in the Cancers. The submitted paper is well written and logically constructed. Therefore, I would like to recommend this manuscript for the publication.

Author Response

We thank the reviewer for his/her kind comments.

Reviewer 2 Report

Ciccarone et al. Epigenetic insights on PARP-1 activity in cancer therapy is very nice review highlighting the role of PARP-1 in DNA/histone epigenetic modifications in addition to more well characterized role in DNA repair.  The putative connections to PARPi resistance and potential role of epigenetic drugs in combatting resistance are discussed. This is very interesting and relevant topic and the manuscript is well written providing extensive detailed information covering most of the areas relevant for the topic. The mechanism of action of PARPi and the conceptual basis as well as current indications should be more clearly written. Maybe also a figure or table to clarify can be considered. Specific comments:

18-23: Text in these sentences is repetitive and should be rewritten.

65-75: add here the concept of synthetic lethality and original references Farmer et al, Nature 2005; Bryant et al, Nature 2005. Elaborate the known mechanisms of resistance a little bit more.

365: Indications of PARPi should be expanded to include also prostate cancer and pancreatic cancer

The authors should go through clinical.trials.gov databases to get information of ongoing trials studing PARPi and epigenetic drugs and add a summary/highlights

BET inhibitors are considered also to have epigenetic effects – these are studied in context of PARPi and could be added.

Author Response

We thank the reviewer for his/her suggestions that permitted us to improve our manuscript.

Our changes in the manuscript are highlighted in yellow in the PDF manuscript file.

Point-by-point reply

Comment:18-23: Text in these sentences is repetitive and should be rewritten.

Answer: according to this suggestion, we modified the abstract.

Comment: 65-75: add here the concept of synthetic lethality and original references Farmer et al, Nature 2005; Bryant et al, Nature 2005. Elaborate the known mechanisms of resistance a little bit more.

Answer: we have introduced the  concept of synthetic lethality, inseretd original references and elaborated the mechanisms of resistance to PARPi

Comment: 365: Indications of PARPi should be expanded to include also prostate cancer and pancreatic cancer

Answer: we have included prostate and pancreatic cancer in the list of tumours treated with PARPi

Comment: The authors should go through clinical.trials.gov databases to get information on ongoing trials studying PARPi and epigenetic drugs and add a summary/highlights

Answer: we have inserted table 2 describing ongoing clinical trials using PARPi in combination with epigenetic drugs

Comment: BET inhibitors are considered also to have epigenetic effects – these are studied in context of PARPi and could be added.

Answer: we added information on co-treatment of BET inhibitors with PARPi in the conclusion section.